# Characteristics of 3D Printed Biopolymers for Applications in High-Voltage Electrical Insulation

**DOI:** 10.3390/polym15112518

**Published:** 2023-05-30

**Authors:** Robert Sekula, Kirsi Immonen, Sini Metsä-Kortelainen, Maciej Kuniewski, Paweł Zydroń, Tomi Kalpio

**Affiliations:** 1Hitachi Energy Research, ul. Pawia 7, 31-154 Kraków, Poland; 2VTT Technical Research Centre of Finland Ltd., Kivimiehentie 3, FI-02044 Espoo, Finland; kirsi.immonen@vtt.fi (K.I.); sini.metsa-kortelainen@vtt.fi (S.M.-K.); 3Department of Electrical and Power Engineering, Faculty of Electrical Engineering, Automatics, Computer Science, and Biomedical Engineering, AGH University of Krakow, Al. Mickiewicza 30, 30-059 Kraków, Poland; pawel.zydron@agh.edu.pl; 4Brinter, 4c Itäinen Pitkäkatu, 20520 Turku, Finland; tomi.kalpio@brinter.com

**Keywords:** cellulose electrical insulation, 3D printing of biopolymers, large-format printing

## Abstract

Three-dimensional printing technology is constantly developing and has a wide range of applications; one application is electrical insulation, where the standard technology uses polymer-based filaments. Thermosetting materials (epoxy resins, liquid silicone rubbers) are broadly used as electrical insulation in high-voltage products. In power transformers, however, the main solid insulation is based on cellulosic materials (pressboard, crepe paper, wood laminates). There are a vast variety of transformer insulation components that are produced using the wet pulp molding process. This is a labor-intensive, multi-stage process that requires long drying times. In this paper, a new material, microcellulose-doped polymer, and manufacturing concept for transformer insulation components are described. Our research focuses on bio-based polymeric materials with 3D printability functionalities. A number of material formulations were tested and benchmark products were printed. Extensive electrical measurements were performed to compare transformer components manufactured using the traditional process and 3D printed samples. The results are promising but indicate that further research is still required to improve printing quality.

## 1. Introduction

Composite materials are broadly used as electrical insulation in a vast number of electrical products. Epoxy resins, XLPE, polyurethanes and liquid silicon rubbers are mostly popular in such high-voltage (HV) applications due to their excellent dielectric properties (electrical withstand strength, dielectric constant, dielectric loss), high mechanical strength (also at elevated temperatures) and chemical resistance [1,2,3,4]. These materials can also be easily processed using molding (casting) technologies [5] to assure proper shape and the desired functionality of the final product. In most of such thermoset materials, functional fillers are used (e.g., alumina or silica) that improve the mechanical performance of the composite and minimize potential shrinkages during the curing process [6]. The incorporation of filler particles also results in higher thermal conductivity of the composite, thus leading to better heat dissipation from the electrical apparatus [7,8,9].

Cellulose-based formulations are another form of the materials used broadly as HV electrical insulation, and power transformers are the product where, in addition to transformer oil, such functional materials play an important role as the insulating media. Various forms of cellulose materials can be found in transformers, and different forms of materials are produced. Pressboard and presspahn are key materials in the manufacturing of transformer insulation components. They are manufactured using the purest cellulose extracted from long-fibered coniferous trees, additional binding agents and a pressing process. From this process, the obtained pressboard is characterized by high mechanical strength and ability to be impregnated with transformer oil, assuring very good dielectric performance [10,11]. Due to the variety of shapes and sizes required, very often, transformer insulation components are manufactured from wet cellulose pulp using metallic molds.

Additive manufacturing/3D printing is one of the most booming technologies in the last decade; this refers both to printing techniques and a number of available materials. In short, additive manufacturing refers to printing layers of material by adding one layer after another to build the solid object, instead of molding or cutting or bending the materials. Today, additive manufacturing (AM) is disrupting conventional manufacturing based on machining, casting, molding and forming technologies in several mature and traditional industries, e.g., automotive, aerospace, etc. With tremendous advances in AM technologies in recent years, the pace of AM adoption is increasing dramatically within different businesses, thanks to the advantages that AM brings with respect to building complex products, speeding up product innovation, reducing size and weight of the components with reduced waste and enabling new service business models.

The three-dimensional printing of polymeric materials is a mature technology in the area of additive manufacturing, offering a broad portfolio of commercial methods and a vast range of materials that can be processed. In general, printing technologies can be divided into a few categories, some of which are (a) fused deposition modeling (FDM), (b) binder jetting (3DP), (c) photopolymerization, (d) selective laser sintering (SLS) and (e) material jetting (inkjet printing). The photopolymerization technique involves the solidification of photo-sensitive resin by means of a high-energy light and is used in stereolithography (SLA) and digital light processing (DLP) printing methods.

Constant development and research in 3D printing technology and the used materials has enabled the use of printed components as parts of electrical devices. Three-dimensional printing provides pros and cons in the forming of electrical device parts depending on the used technology and the printed part’s application. There have been a number of research investigations aiming to develop printable cellulose. The current industrial technologies have limitations in forming three-dimensional structures from cellulose, which could be beneficial in complexly shaped soft or hard objects. A review of the literature shows the possibilities of using 3D printing to construct parts of electrical motors [12,13] and thermoplastic elements with embedded wires [14].

The authors in [15] report the impact of thermoplastic 3D printing technology on the electrical properties of formed parts as the elements of electrical insulation. The results show that stereolithography apparatus (SLA), fused deposition modeling (FDM) and selective laser sintering (SLS) provide parts with similar permittivity but different dissipation factors and volume resistivity; the measured values of dielectric withstand of the printed parts depend on the technology used. The selection of the proper technology and the proper material should be made individually according to the application of the printed part. The authors of [16] report changes in dielectric strength due to different printing directions. The results of [17] show the impact of the used filament type in FDM printing on the permittivity, dissipation factor and dielectric strength. The authors considered the impacts of different printing resolutions in a range of 50 µm to 200 µm.

Three-dimensional printing is used to make high-voltage insulators from dielectric functionally graded materials [18]; the authors analyzed the impact of the concentration of TiO_2_ on the relative permittivity and flashover voltage. The developed insulators have up to 17.5% higher flashover voltage than standard ones. In [19], the authors used stereolithographic 3D printing based on epoxy doped with alumina composite in the forming of HV insulating spacers.

There have been a number of research investigations aiming to develop printable cellulose. The current industrial technologies have limitations in forming three-dimensional structures from cellulose, which could be beneficial in complexly shaped soft or hard objects. Three-dimensional printing is a technology that can be used to form complex structures by laying down successive material layers until the object is complete [20]. Three-dimensional printing also enables the mass customization of products on-demand [21]. Increased awareness of environmental aspects has raised interest in the replacement of oil-based synthetic polymers in 3D printing with their bio-based renewable counterparts, including cellulose [22,23]. However, the utilization of cellulose in 3D-printing is challenging due to the inherent properties of cellulose materials [24]. Pure cellulose is not a thermoplastic material, which prevents its direct utilization in fused deposition modelling (FDM), which is the most-used 3D-printing technology utilizing polymeric materials. When cellulose is chemically grafted with suitable side chains, typically using esterification reactions, the feature of thermoplasticity can be added [25], making it completely processable at 200 °C without the addition of a plasticizer [26,27,28], even in 3D printing. Another suitable method to 3D-print cellulose is direct writing, which utilizes paste-like materials dissolved or dispersed in a liquid solvent. This approach has been successfully employed with cellulose directly dissolved into ionic liquids for printing spatially tailored 3D gels and membrane structures [29].

An analysis of the usage of biodegradable polymer nanocomposites in the printing of electronic structures is presented in [30], where the researchers used carbon-black-nanoparticle-embedded PLA as an environmentally friendly material in the printing of FRID tags. The authors of [31,32] present a review of the possibilities of cellulose material usage in 3D printing in the fields of biomedical and smart health care, printed electronics and responsive wearable textiles.

When discussing the requirements of cellulose materials used as transformer insulation in particular, it has been concluded that there is no suitable printable material, and such a new formulation would need more research. The results of such investigations are presented in this paper.

## 2. Molded Cellulose Insulation in Power Transformer Application

A number of molded components can be found in a power transformer. These include various types of rings, snouts, collars and others. Selected sample products are presented in Figure 1.

These insulation components are produced using special processing procedures, which guarantee high compaction and dimensional stability resulting in the required durability and reliability in case of short-circuit events. The manufacturing process is carried out using metal molds (Figure 2).

This wet pulp process is complex, multi-stage, long-lasting, energy-consuming, laborious and can result in significant material waste of up to 60%. It comprises a great number of steps: designing the transformer and insulation components, designing metal molds and their manufacturing, preparing the wet cellulose sheet, forming the wet cellulose sheet on the metal mold, compressing the wet cellulose sheet, drying the wet cellulose sheet in a convective oven or using an electric drying process depending on the component to be dried, removing the dried component from the metal mold and, finally, machining the dried component to obtain the final shape of the component.

Each power transformer unit usually has a specific and individual design, which also influences the design of the insulation components. It means, in practice, that for each insulation component, the geometry of an individual metal mold has to be designed and manufactured.

Having in mind the above-mentioned challenges accompanying wet cellulose molding, in the frame of the EU funded project NOVUM (European Commission Horizon 2020/SPIRE, proposal number: 768604, proposal acronym: NOVUM), a new functional biobased material was developed that is suitable for 3D printing technology. Such a new additive manufacturing-based process offers a number of benefits compared to existing manufacturing: a number of process steps can be eliminated, no molds are required, much less waste is generated, and the energy-consuming drying of wet pulp is not necessary.

## 3. Development of New Material

### 3.1. Requirements for New Materials

Dozens of cellulose-based material formulations have been developed in various forms as pastes, filaments and pellets, and they have been widely tested. It is obvious that such a biomaterial should fulfill a number of technical requirements, and a proper material formulation had to be developed from scratch, since there were no available commercial materials. Most of such commercial printable materials show too-weak mechanical performance, especially at elevated temperatures, and their dielectric performance is too weak for high-voltage applications. There were also trials for the printing of pure cellulose, but the mechanical withstand of such prints was questionable.

A new material to be used in the 3D printing of electrical insulation components should be, first of all, oil-compatible, and also must be characterized by high electrical withstand strength (above 20 kV/mm). In addition to the required mechanical properties, the only possible material seemed to be a moldable polymeric one with suitable functional additives.

Cellulose as such is a polymeric material achieved from various natural raw materials [33], but it is not moldable without chemical modification. In order to make it thermally moldable, it needs to be modified by esterification or etherification; these are the most common commercial methods in use [34,35]. Cellulose-based thermally moldable films and injection moldable materials in ester form such as cellulose acetate (CA), cellulose acetate butyrate (CAB) and cellulose acetate propionate (CAP) are commercially available for several industrial uses. The longer C-chain derivatized cellulose chains, such as esterification with propionic acid, provide lower processing temperatures in thermoplastic converting processes, enabling also the introduction of neat cellulose in the developed material.

Extrusion-based 3D printing of wood cellulose in a long fiber form can be challenging due to small nozzles that can be potentially clogged due to fiber agglomerates. For this reason, microcellulose powder was selected as a potential cellulose-based filler for the composite material. Suitable powders are provided, for example, by JRS Rettenmeier [36].

Cellulose in particle form, for example microcellulose, being very hydrophilic, typically needs some added compatibilizer for hydrophobic thermoplastic materials. In our approach, the compatibilizer needed to be from renewable origins and able to couple fiber-form cellulosic with cellulose ester, CAP. Epoxy functional natural oils, such as epoxy functional linseed oil, were assumed to be suitable for this purpose due to previous experience with cellulose fiber polylactic acid (PLA) composites [37].

### 3.2. Materials and Compound Manufacturing

The compound used in this study was similar to that presented previously in another article by the authors [38]. The materials used for manufacturing the cellulose-based compound were cellulose acetate propionate (CAP) (CELLIDOR CP300-13, Albis Plastics GmbH, Hamburg, Germany) with a phthalate-free plasticizer content of 13% and a melt flow rate of 7.5 cm^3^/10 min (210 °C, 2.16 kg) as the polymer matrix, 20 wt.% microcrystalline cellulose (VIVAPUR 105, JRS Pharma GmbH, Weissenborn, Germany) with an average particle size based on laser diffraction of 15 µm as the cellulosic filler, and reactive epoxidized linseed oil (Lankroflex™ L, Valtris Specialty Chemicals, Independence, OH, USA) as the coupling agent.

The microcellulose (MC) fibers were treated with the Lankroflex L coupling agent before compounding by mixing 5% Lankroflex L with MC in relation to MC dry weight using a blade blender. The MC-blend was dried overnight in a heat convection oven at 50 °C. The polymer CAP was dried at 80 °C for 2 h before compounding. The MC-blend was compounded with CAP using a co-rotating twin-screw extruder (Berstorff ZE 25 × 33 D, Berstorff GmbH, Hanover, Germany). The extruder zone temperatures ranged from 80 to 205 °C, the speed was 100 rpm and material output 2 kg/h. The small granules produced from compounding were used for both injection molding to test the material properties and extrusion 3D printing.

### 3.3. Injection Molding

Injection molding to prepare the test specimens in line with ISO 527 was performed with an injection molding machine (Engel ES 200/50 HL, Engel Maschinenbau Geschellschaft m.b.H, Schwefberg, Austria). The injection molding process temperatures in the screw from feed to screw were from 180 to 200 °C and the mold temperature was 70 °C.

### 3.4. Testing of Injection-Molded Test Bars

Mechanical properties such as tensile properties, impact strength and thermal property heat distortion temperature (HDT) were analyzed from injection-molded standard test bars. The tensile strength was tested according to the ISO-527 standard using an Instron 4505 Universal Tensile Tester (Instron Corp., Canton, MA, USA) and an Instron 2665 Series High-Resolution Digital Automatic Extensometer (Instron Corp., Canton, MA, USA) with a 10 kN load cell, a 5 mm/min crosshead speed and five parallel specimens.

The Charpy Impact strength was measured according to the ISO-179 standard using a Charpy Ceast Resil 5.5 Impact Strength Machine (CEAST S.p.a., Turin, Italy) for six to ten parallel unnotched specimens.

The ISO-75 standard was used to measure HDT for the samples using method A with 1.8 MPa stress on the samples. The equipment for HDT measurement was the Ceast HDT 3 VICAT P/N 6911.000 (Ceast S.p.a., Turin, Italy) system with three parallel samples.

### 3.5. Results for the Cellulose-Based Compound

The test results of the material properties of injection-molded specimens tested at VTT are presented in Table 1.

## 4. Material Testing and Characterization

### 4.1. Oil Compatibility

The materials used inside power transformer tanks, reactors and components that will come in contact with transformer oil must be compatible with the oil, i.e., the material must neither adversely affect the properties of the oil nor be degraded by the oil.

Before the material was placed in the exposure vessel, it was rinsed with a portion of the prepared oil, which was discarded. The material was then placed in the vessel and oil was added in such an amount that the relation between panel surface area/weight and oil volume became 10 times higher than that under service conditions. The vessels were covered with aluminum foil and then exposed to 90 °C for 7 days (Figure 3).

The evaluation of the exposed oil was performed based on visual assessment, measurement of dielectric dissipation factor according to IEC 60247 [39], interfacial tension against water according to ASTM D971-99A [40], and increased gas production according to IEC 60567 [41].

In general, any compounds that were released from the material during the test listed in the table above or those that had properties of polar and/or dipole compounds (acids, aldehydes, ketones, ionic compounds, etc.) may have negatively affected the compatibility test results.

The material under testing after aging in the oil was evaluated visually, inspected by touch and compared with the as-delivered samples. The appearance and color of the oil as delivered and after the test with the material and without its presence were evaluated visually.

Both the dissipation coefficient tan δ and the water–oil interfacial tension had the greatest influence on the deterioration of these parameters by polar compounds. The transformer oil is a non-polar substance, so it has good insulating properties. In the case of introduction of polar and/or dipole compounds (acids, aldehydes, ketones, ionic compounds, etc.), the interfacial tension would decrease, as there would be compounds having a higher affinity for water.

Based on the performed experiments, the newly developed material exhibits a good oil compatibility and can be used as oil-filled transformer insulation.

### 4.2. Dielectric Properties

High-voltage electrical insulation components must fulfill the requirements of optimal dielectric properties to achieve the best performance of the insulation in a long-term service. This requirement depends on the application of power equipment, its nominal ratings and local company regulations. There are several electrical properties that should be checked before the application of the selected material as a component of the insulation system. The most important dielectric properties are as follows: surface and volume resistivity, permittivity, dissipation factor, partial discharge and dielectric withstand. These properties determine the electrical field distribution in different operating conditions, power losses, the aging process and the maximal operational electrical field, which implies the voltage levels. The methodology of the measurement of these selected dielectric properties is standardized by the relevant IEC committees [42,43,44,45,46,47,48,49,50,51,52] or local national standardizing committees.

The investigations of new types of materials should be performed on basic test samples in the form of flat samples. This kind of shape provides a uniform electrical field distribution in the tested material’s volume. Determination of the (1) volume and (2) surface resistivity of the flat solid dielectric materials was made in accordance with the IEC 62631-3-1, IEC 62631-3-2 and IEC 61340-2-3 standards [44,45,46]. The test stand used in the investigations consisted of the electrometer KEYSIGHT B2987A and Resistivity Cell (16008B) (Figure 4) equipped with normalized electrodes [47]. The electrometer could provide DC voltage up to ±1000 V; this voltage was used in the test to provide the electrical field in the test sample in the range of 1 kV/mm. The measured resistance was converted to volume or surface resistivity (Figure 4). The duration of the measurements was 180 s; it was assumed that after this time the polarization processes were negligible.
(1)$$\rho_{v}{=R}_{x}{{(d}_{1}+g)}^{2}\times \frac{\pi }{4h}$$
(2)$$\rho_{s}{=R}_{x}{(d}_{1}+g)\times \frac{\pi }{g}$$
where:

ρ_s_: surface resistivity, [Ω];

ρ_v_: volume resistivity, [Ωm];

R_x_: measured surface or volume resistance, [Ω];

d_1_: diameter of inner electrode, [m];

g: gap between electrode and guard ring, [m];

h: thickness of the sample, [m].

The investigations also covered some non-standardized testing procedures with the frequency dielectric spectroscopy (FDS) method. This method is used for permittivity and dissipation factor determination as a function of frequency. The laboratory stand was equipped with a Frequency Response Analyzer Solartron 1260 with a 1296 dielectric interface (Figure 5). This system is suitable for oil–cellulose material characterization in comparative tests. The measurements for the sample’s characterization were performed for frequencies in the range of 10 mHz to 1 kHz. The test voltage was set to 3 Vrms.

The second group of dielectric measurements were the destructive test, which aimed to determine the dielectric withstand and partial discharge presence. The tests were carried out in accordance with IEC and ASTM standards (IEC 60243-1 [50], D 149-09 [51]). The testing specimen consisted of two profiled stainless-steel electrodes (with diameters 25 mm/75 mm) (Figure 6). The slow-rate-of-rise voltage method was applied. The test stand consisted of a Voltage Test Transformer TP60 with a maximal voltage rating of up to 60 kV, a Phoenix KiloVolt meter and test electrodes. Due to the high value of breakdown, the voltage test samples with the electrode system were placed in insulation oil. Before breakdown, voltage partial discharge measurements were performed.

The preparation of the samples is a key element in the process of determining dielectric properties. A proper procedure minimizes the error resulting from the moisture content, which might stay in a sample, or other issues such as an unfinished curing process. The samples were conditioned in a vacuum of 100 mBar at 80 °C for 12 h. The oil used in the measurements was filtered and conditioned with the same conditions. The samples after conditioning were immersed in oil and put in the vacuum for 12 h.

The identification of partial discharge phenomena (PD) in a new technology and new materials and the level of partial discharge inception voltage (U_PDIV_) can decide the applicability of the analyzed case as a high-voltage insulation component. Test procedures for partial discharge (PD) detection use conventional methods based on the apparent charge measurement, utilizing quasi-integrating detection circuits, as defined in the IEC 60270 standard [48]. The laboratory stands had specialized measuring systems (ICM System, Power Diagnostix) that met the requirements defined in the standard used for PD measurements. As a HV source, the test transformer TP60 was used, and the 75–75 mm electrode system was placed in the insulation oil (Figure 7). This kind of electrode provides a uniform electrical field in the whole area of the tested sample, maximizing the probability of detecting the imperfections causing the PDs. The test procedure covered a range of voltages, with the voltage slowly rising up to the voltage at which partial discharges occurred (U_PDIV_). The maximal PD test voltage limit was equal to an electrical field of 15 kV/mm; this level was set to prevent electrical breakdown in the sample. If no PD occurred up to this voltage level, the sample was assumed to be PD free. The recorded background noise levels in the test stand did not exceed 0.5 pC at 20 kV.

In the nondestructive evaluation of the developed material, two sets of samples were measured; in the first set, raw samples were used, while in the second set, samples after conditioning and oil impregnation process were measured.

This section presents the measurement results of selected dielectric properties (volume resistivity, permittivity, dissipation factor, PDs, and breakdown voltage) for the biobased test material labeled N28.

The ambient conditions during measurement were as follows: temperature 21 °C, air humidity 24%, pressure 999 hPa. Two sets of samples were tested, first raw material samples without oil and before conditioning, then conditioned samples immersed in oil. The measurements of the dielectric properties of the raw materials gives information on the developed material and used technology. The conditioned samples imitate the target application of the developed material.

The volume resistivity results are presented in the figures below. Figure 8 presents the volume resistivity time characteristics of the raw samples before the conditioning process. Figure 9 presents a set of 5 samples conditioned and immersed in oil.

The quasi-steady-state volume resistivity values of the raw samples and conditioned samples are presented in Table 2. For conditioned samples, the average value of volume resistivity is presented.

An analysis of the presented results shows that the characteristics of raw and conditioned samples of the proposed bio-based material N28 for HV insulation components have stable and repeatable values, with steady-state resistivity at a similar level for every conditioned sample (Table 2). The standard deviation is 1.53 × 10^13^ Ωm, which is equal to a 3% deviation. There is a noticeable difference between the raw and conditioned material, but the level of resistivity places this material as a fair insulator. The reason for the conditioning process increasing the resistivity might be that during conditioning, the cellulose filler dries; additionally, the polymeric matrices harden and crosslink.

Figure 10 and Figure 11 present the relative permittivity and dissipation factor characteristics of the raw and conditioned samples. The summary of the averaged results for 50 Hz are presented in Table 3. As can be seen, the raw material has a higher permittivity and dissipation factor than the conditioned samples. Taking into consideration the results of volume resistivity and the characteristics of permittivity and dissipation factor, it can be concluded that the raw sample has moisture in its cellulose filler. Thus, it should be conditioned before application as part of a high-voltage insulation system. The conditioning process removes bound water, the dielectric constant of which is approx. 80; this is reflected in the lowering of the permittivity, especially in the low-frequency region. During conditioning, the maximum of the dissipation factor moves to the lower frequencies, which is also related to the removal of moisture from cellulose. The thickness of the materials is not uniform, thus there is a visible dispersion of results. The repeatability between conditioned samples is fair, the maximal difference between dielectric constant values is 2%, and the difference between dissipation factors is 4%.

The last classification of dielectric properties for the N28 material was to check for the occurrence of partial discharges and determine the breakdown voltage, which is the basis for dielectric strength calculation. Partial discharges are unpleasant phenomena because they speed up the aging process of the insulation system, provide additional power losses and cause radio disturbances. In the worst case, they can cause damage to the insulation before the assumed time of life. The origins of partial discharges can be different; most commonly, they are air voids in solid dielectrics or metallic intrusions provided during the manufacturing process. The measurements were performed on conditioned samples immersed in oil.

The results presented in Table 4 show the breakdown strength and partial discharges inception voltage of the N28 samples. The mean electrical breakdown strength of NOVUM 28 material is 26.7 kV/mm, which is above the assumed limit of 20 kV/mm. Partial discharges are not observed up to 15 kV. The manufacturing process applied to the flat samples measured for their electrical properties was mold injection; as can be seen in the PD results, this process did not provide any air voids or metallic intrusions to the test material.

### 4.3. Benchmark versus Printed Product

The benchmark samples were manufactured at Hitachi Energy Insulation Kit Center in Lodz, Poland using traditional wet-pulp cellulose.

The components to be compared were printed by Brinter in Turku, Finland using the printer developed in the course of the NOVUM project (Figure 12).

Both components are presented in Figure 13.

A number of printing trials were performed using various printing parameters, printing speeds (50, 10, 200, and 300 mm/s), and extrusion nozzles with various diameters (1, 3, and 5 mm). The best printing quality was obtained for the following set of printing parameters:Nozzle diameter, 1 mm;Layer height, 0.35 mm;Tool temperature, 179/206/219 °C (top/middle/nozzle);Printing speed, 50 mm/s;Printing bed temperature of 40–45 °C with ABS coating;Printing time, 6 h.

The final assessment of the applicability of N28 material in 3D printed HV insulation components was based on the comparison of the dielectric properties of the conventional water-formed component and the 3D printed component. The results comparison is presented below. Figure 14 presents the comparison of volume resistivity characteristics for the standard technology and the NOVUM printed sample; a summary of the results is presented in Table 5.

The characteristics presented in Figure 14 show that the standard technology samples have at least one order of resistivity less than the N28 printed samples. Both samples are of the same class of current conduction ability. Figure 15 and Figure 16 present the permittivity and dissipation factor wide-band spectrum characteristics. A summary of the results specifically at 50 Hz is presented in Table 6.

The characteristics presented in this section show high similarity between both materials. The standard technology samples have a dielectric constant of 3.04 and the N28 samples have a dielectric constant of 3.48, while the dissipation factors are 0.0077 and 0.0064, respectively.

The last comparison focuses on the partial discharge phenomenon and dielectric breakdown field. The test setups are presented in Figure 17 and Figure 18. The final products have two kinds of surfaces; (Figure 17) shows the determination of the breakdown voltage using flat surfaces, while Figure 18 shows the determination of the breakdown voltage using curved surfaces. Both measurements were standardized according to the IEC.

The results presented in Table 7 and Table 8 show the breakdown strength and partial discharge inception voltage values of the characterized samples for flat and curved surfaces. The breakdown voltage for the standard-technology samples exceeded the limits of the equipment and the test stand electrodes, while the N28 technology samples had significantly lower breakdown strength (mean 13.57 kV/mm for flat surfaces). The breakdown strength for the standard-technology samples was almost 2 times higher (>23.4 kV/mm). The curved surfaces of the N28 technology samples were characterized by a mean breakdown field in the range of 15.62 kV/mm, while the standard-technology samples had a breakdown field for curved surfaces higher than 23 kV/mm. The measurement results of partial discharges shows that the standard-technology samples were mostly without partial discharges in the analyzed voltage range, while the N28 technology samples had partial discharges starting at a mean level of 33 kV, which is comparable to the breakdown voltage of the samples. The resolved patterns of the partial discharge phase measured for these samples are similar to the patterns of gas void defects inside the material (Figure 19). Polymeric matrices and 3D printing technology can provide voids in the printed structure that are impossible to remove. The refinement of this technology is a next step.

Figure 20 presents the visible imperfections of the 3D printed sample using the bio-based polymer. This technology provides voids at curved surfaces, which might be problematic due to the lower dielectric withstand of these areas and partial discharge phenomena. Additionally, the non-uniform structure might be problematic, with the development of a discharge process at interfaces.

## 5. Conclusions

Based on the obtained results, we found that the printed material still does not fulfill the requirements for materials used as transformer insulation. Our results indicate that at the current stage of development, material exhibits much better mechanical and dielectric properties when injection molding is used instead of 3D printing. Most likely due to internal voids resulting from the printing process, the structure of the material is not fully compact and such properties as electrical breakdown strength are affected.

On the other hand, basic dielectric properties such as the permittivity and dissipation factor of the printed material are suitable for transformer insulation, which confirms that this material formulation is very promising and can bring a number of sustainable benefits (less energy for drying, minimized waste and design freedom for better functionality). However, more deep investigations of printing optimization are required to assure better interface quality without the appearance of internal voids in the printed components.

## Figures and Tables

**Figure 1 polymers-15-02518-f001:**
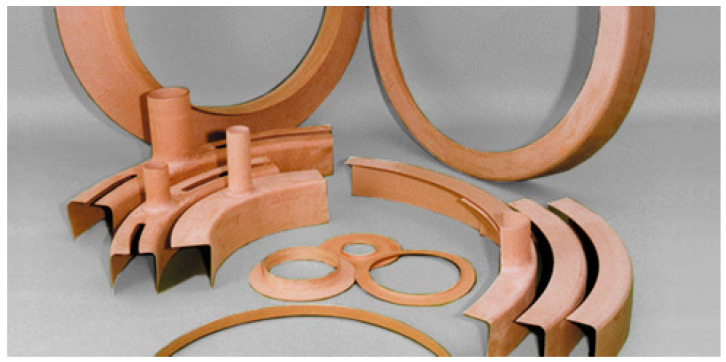
Samples of transformer cellulose components.

**Figure 2 polymers-15-02518-f002:**
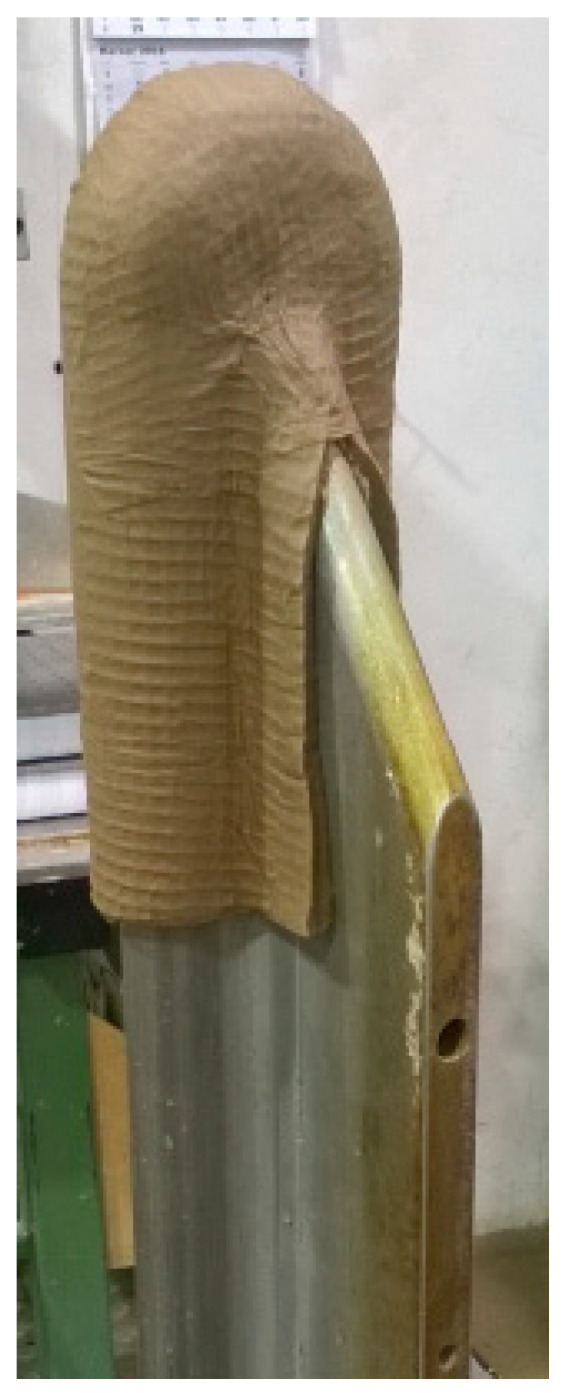
Molding of wet cellulose pulp.

**Figure 3 polymers-15-02518-f003:**
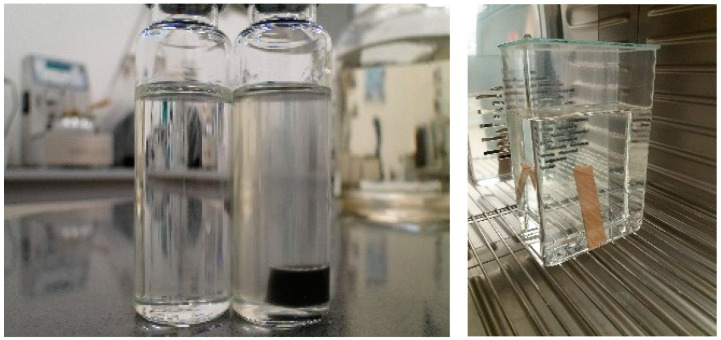
Physico-chemical compatibility (7 days, 90 °C).

**Figure 4 polymers-15-02518-f004:**
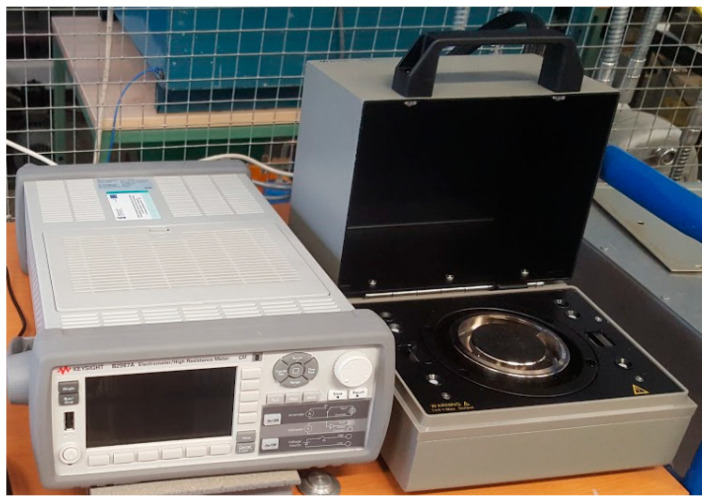
Electrometer with Resistivity Cell.

**Figure 5 polymers-15-02518-f005:**
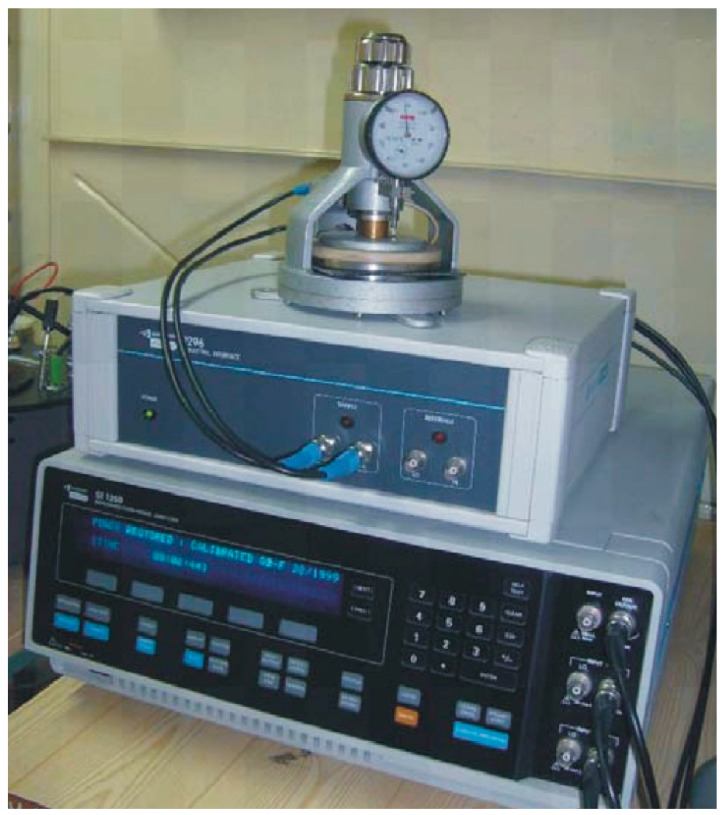
Frequency Response Analyzer Solartron 1260 with dielectric interface 1296 for frequency dielectric spectroscopy characterization of dielectric materials.

**Figure 6 polymers-15-02518-f006:**
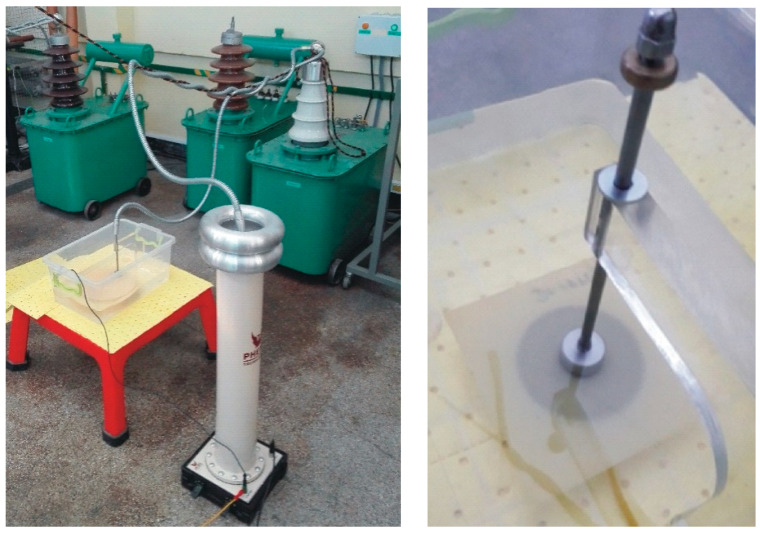
Example of breakdown voltage testing setup (**left**) and insulating material sample between standard testing electrodes in oil (**right**).

**Figure 7 polymers-15-02518-f007:**
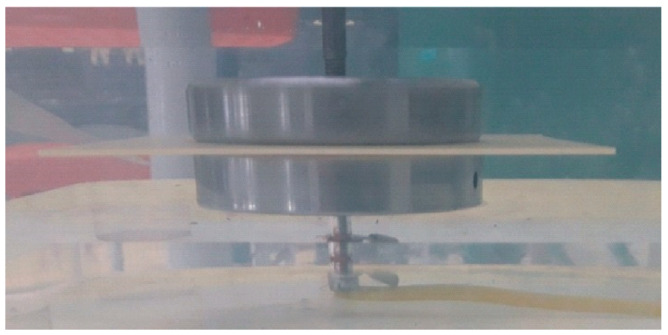
Material sample (in oil) between profiled stainless-steel electrodes with equal 75 mm diameters.

**Figure 8 polymers-15-02518-f008:**
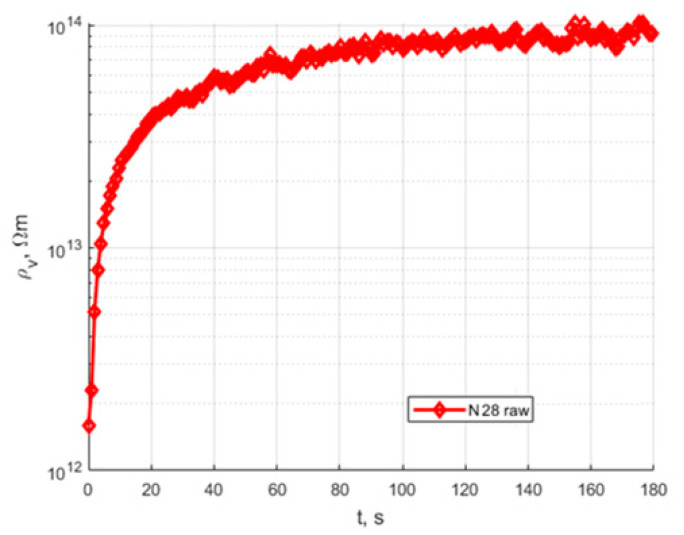
Time characteristics of volume resistivity of N28 sample before conditioning process.

**Figure 9 polymers-15-02518-f009:**
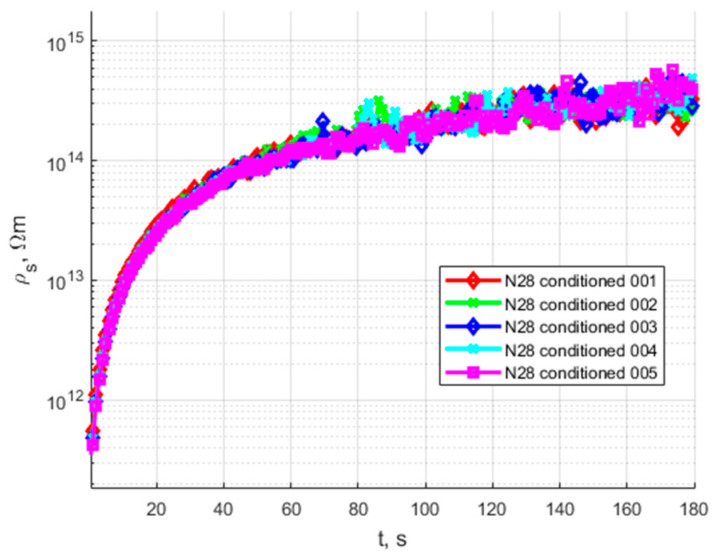
Time characteristics of volume resistivity of conditioned and oil-impregnated N28.

**Figure 10 polymers-15-02518-f010:**
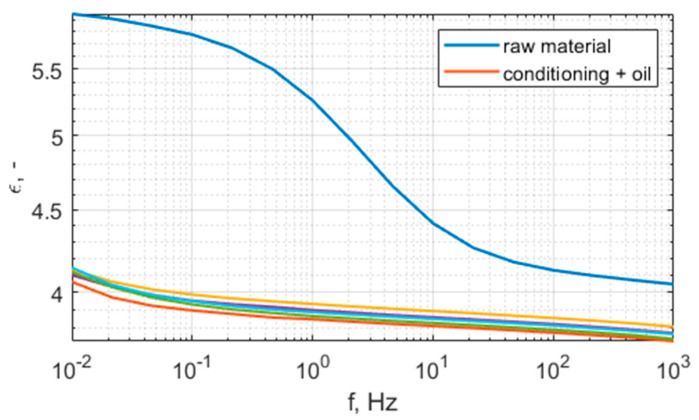
Relative permittivity spectrum of N28 material, raw material (blue), conditioned (5 samples plotted with different colors).

**Figure 11 polymers-15-02518-f011:**
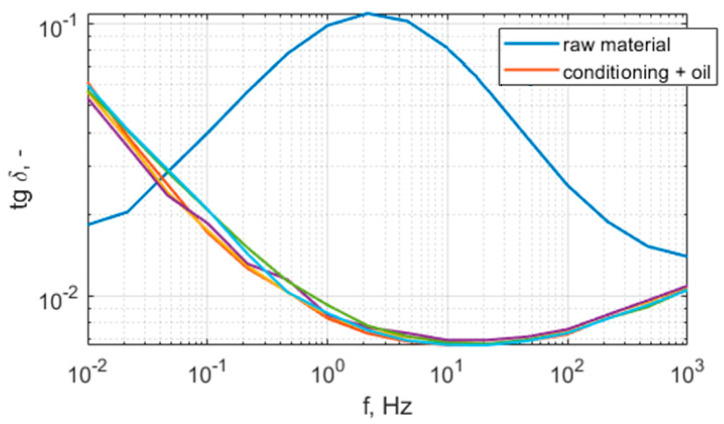
Dissipation factor spectrum of N28 material, raw material (blue), conditioned (5 samples plotted with different colors).

**Figure 12 polymers-15-02518-f012:**
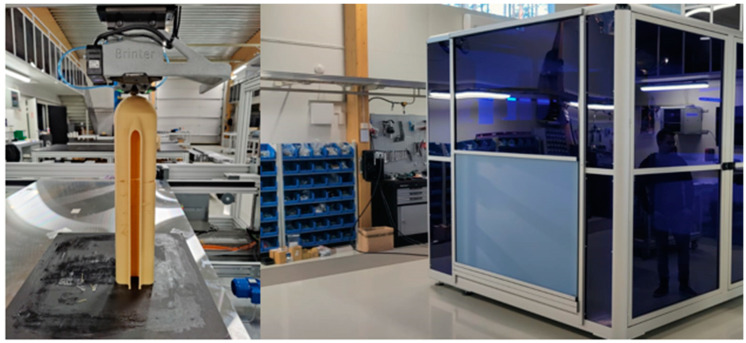
NOVUM large-volume-components 3D printer.

**Figure 13 polymers-15-02518-f013:**
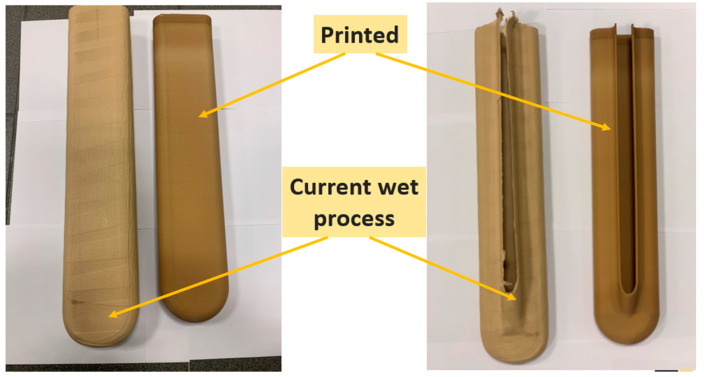
Components being analyzed.

**Figure 14 polymers-15-02518-f014:**
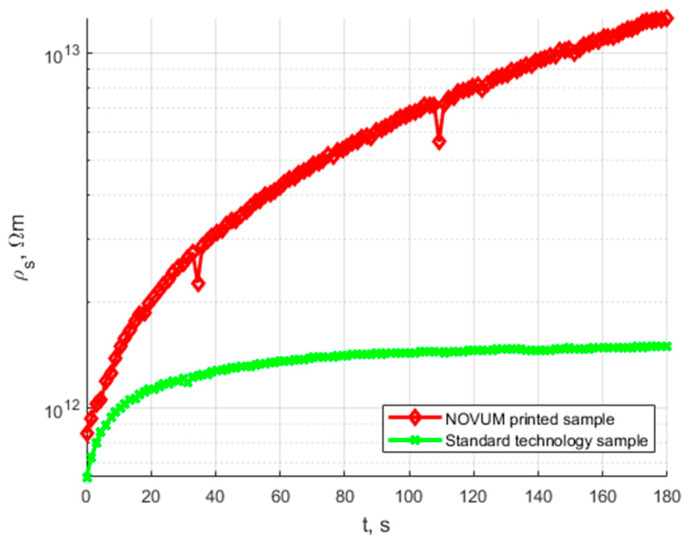
Time characteristics of volume resistivity of N28 printed sample and standard technology sample.

**Figure 15 polymers-15-02518-f015:**
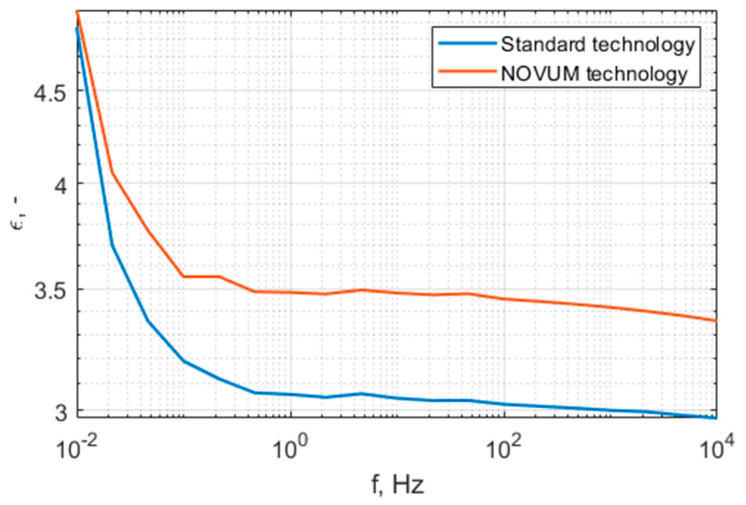
Permittivity spectrum for standard-technology and N28 technology materials.

**Figure 16 polymers-15-02518-f016:**
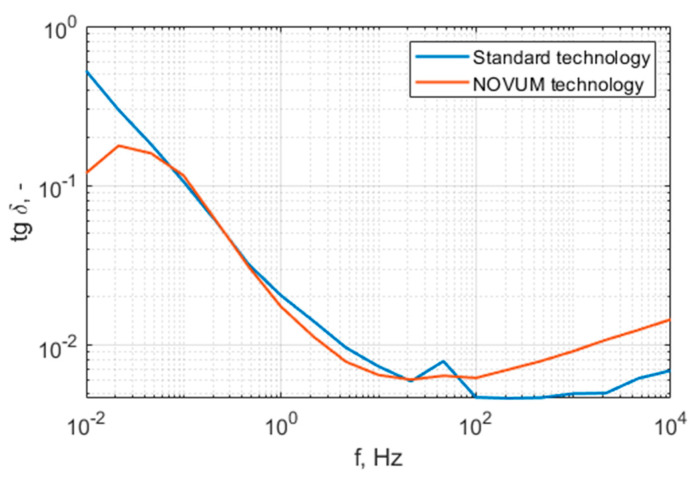
Dissipation factor for standard-technology and N28 technology materials.

**Figure 17 polymers-15-02518-f017:**
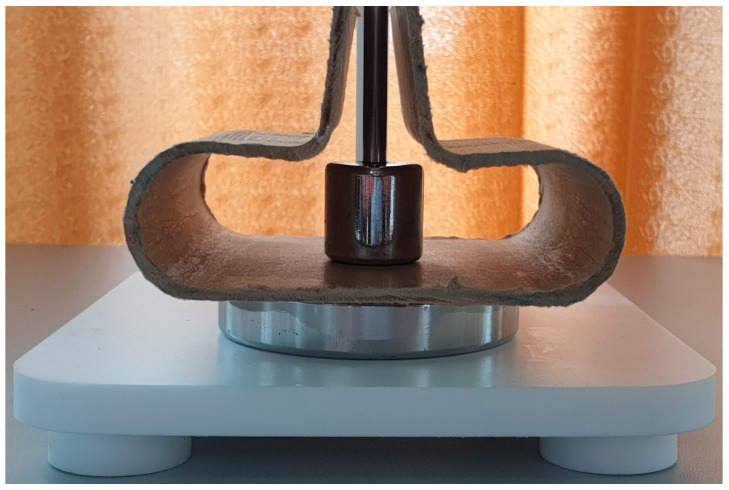
The test sample between cylinder 25 and 75 mm electrodes used for testing of flat surfaces.

**Figure 18 polymers-15-02518-f018:**
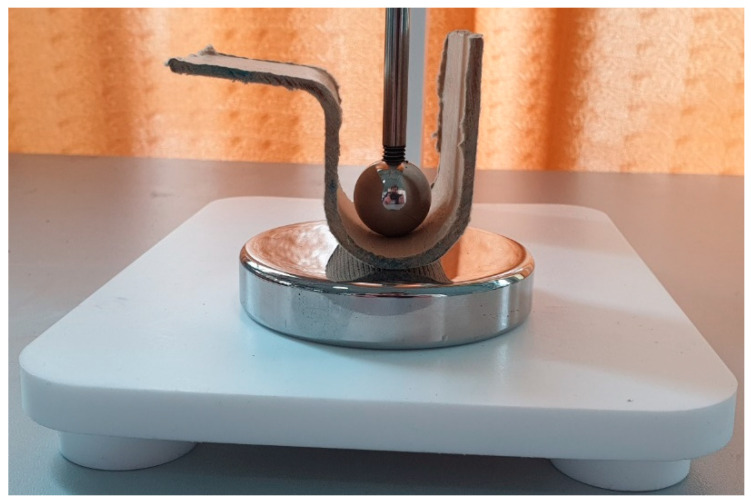
The test sample between sphere 20 mm and cylinder 75 mm electrodes used for testing of curved surfaces.

**Figure 19 polymers-15-02518-f019:**
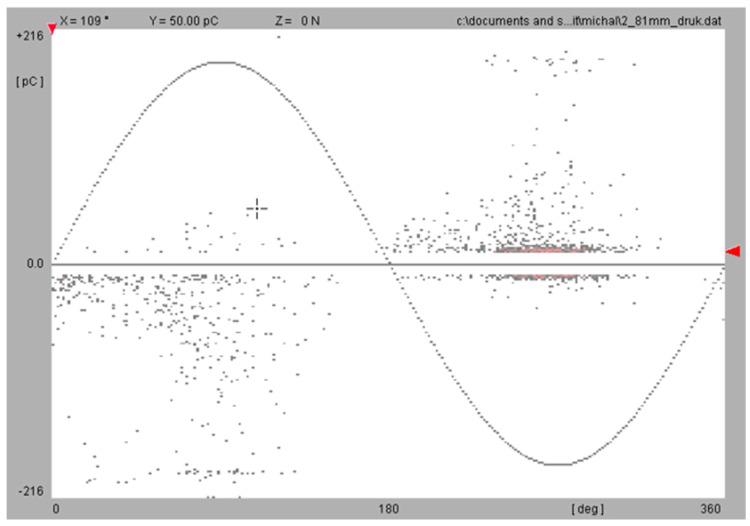
Exemplary PDPR pattern measured for NOVUM Technology sample at PD inception voltage.

**Figure 20 polymers-15-02518-f020:**
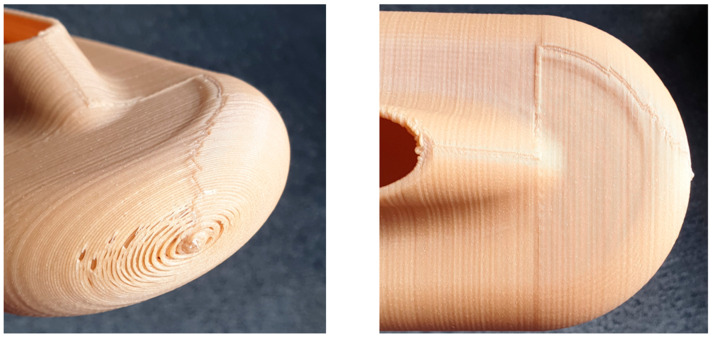
Visible voids and non-uniform structure of N28 3D printed samples.

**Table 1 polymers-15-02518-t001:** Material properties of the samples manufactured by injection molding.

Analysis	Value	Method
Tensile strength at yield, MPa	29.3 ± 0.2	ISO 527
Tensile modulus (Young), MPa	1850 ± 180	ISO 527
Strain at break, %	9.9 ± 0.5	ISO 527
Charpy impact strength (unnotched), kJ/m^2^	21.7 ± 2.6	ISO 179
HDT, °C	60.7 ± 0.5	ISO 75

**Table 2 polymers-15-02518-t002:** Steady-state volume resistivity of N28 sample before conditioning process.

Material	Volume Resistivity
N28 raw	9.62 × 10^13^
N28 conditioned	4.74 × 10^14^

**Table 3 polymers-15-02518-t003:** Comparison of permittivity and dissipation factor measurement data at 50 Hz for N28.

Material	ε_r_ at 50 Hz	tgδ at 50 Hz
N28 raw	4.17	0.0373
N28 conditioned	3.82	0.0070

**Table 4 polymers-15-02518-t004:** Measurement results of partial discharge inception voltage, breakdown voltage and dielectric field withstand of N28.

Value	PDIV	U, kV	h, mm	kV/mm
Average	>15 kV	28.75	1.10	26.66
Std. deviation	-	2.12	0.04	2.20

**Table 5 polymers-15-02518-t005:** Steady-state volume resistivity of sampled before conditioning process.

Tested Material	Volume Resistivity
Standard technology sample	1.49 × 10^12^
N28 printed sample	2.2 × 10^13^

**Table 6 polymers-15-02518-t006:** Comparison of permittivity and dissipation factor measurements for 50 Hz.

Tested Material	Dielectric Constant 50 Hz	Dissipation Factor 50 Hz
Standard-technology sample	3.04	0.0077
N28 printed sample	3.48	0.0064

**Table 7 polymers-15-02518-t007:** Measurement results of partial discharge inception voltage, breakdown voltage and dielectric withstand field for flat surfaces.

Sample	PDIV, kV	BDV, kV	h, mm	E_BDV_, kV/mm
Standard-technology sample set	PD free	>60	2.57	>23.34
N28 printed sample set	33.67	35.67	2.63	13.57
Standard deviation	3.21	1.44	0.15	1.27

**Table 8 polymers-15-02518-t008:** Measurement results of breakdown voltage and dielectric withstand field for curved surfaces.

Sample	BDV, kV	h, mm	E_BDV_, kV/mm
Standard-technology sample set	>60	2.57	>23.38
N28 printed sample set	41.6	2.66	15.62
Standard deviation	4.43	0.08	1.24

## Data Availability

Not applicable.

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
