# Peer review of "Characteristics of 3D Printed Biopolymers for Applications in High-Voltage Electrical Insulation"

_polymers, 2023, doi:10.3390/polym15112518_

Round 1
Reviewer 1 Report
The authors described a 3D printing concept of transformer insulation components using bio polymers. This work is well performed. I have only a few comments for authors to addresses.
1. Figure 14, there are two abrupt data points that away from the curve, which is quite confusing. Why is that?
2. In order to give a bigger picture to readers, the advantages of additive manufacturing need more discussion in the introduction. In addition, different type of 3d printing such as DLP, SLA etc. can be introduced. Besides, the following recent important papers additive manufacturing/3D printing of polymer composites (https://doi.org/10.1039/D1PY00705J; https://doi.org/10.1039/D1PY01283E) should be cited.
3. Figure 11, some of the blue curve is covered by the box. Please revise it.
4. Reference could be added for “ Cellulose in particle form, for example microcellose, and being very hydrophilic needs typically some compatibilizer with hydrophobic thermoplastic materials.”
5. In the abstract, please be specific what “new materials” and what “manufacturing concept” the authors used.
Reviewer 2 Report
In this study, a new material and the 3D printing technology of transformer insulation components were developed, which shows very good application potential. Before publication, some questions should be clarified.
1. During measurement, the current declines with time due to the attenuated absorption current. In Fig.8, the volume conductivity of N28 sample should be about 1e14 Ω/m at t>180s, instead of the average value of the entire process.
2. What are the advantages of the 3D printed components over those under current wet process? From the test results it can be seen that the current cellulose components have much better insulation performances than the printed ones.
3. After resin impregnation, the insulation performances of the 3D printed components might be improved.
